# MET Oncogene Controls Invasive Growth by Coupling with NMDA Receptor

**DOI:** 10.3390/cancers14184408

**Published:** 2022-09-11

**Authors:** Simona Gallo, Annapia Vitacolonna, Paolo Comoglio, Tiziana Crepaldi

**Affiliations:** 1Department of Oncology, University of Turin, Strada Provinciale 142, 10060 Candiolo, Italy; 2Candiolo Cancer Institute, FPO-IRCCS, Strada Provinciale 142, 10060 Candiolo, Italy; 3IFOM, FIRC Institute for Molecular Oncology, Via Adamello 16, 20139 Milano, Italy

**Keywords:** MET tyrosine kinase receptor, hepatocyte growth factor, glutamate receptor, tumor invasion

## Abstract

**Simple Summary:**

The MET oncogene, encoding the tyrosine kinase receptor for a hepatocyte growth factor (HGF), plays a key role in the onset and progression of aggressive forms of breast cancer. Recently, it was found that the glutamate receptor, which has a well-known role in the nervous system, is expressed in many types of tumors outside the nervous system and contributes to metastatic behavior in breast cancer cells. Here, we highlight that MET protein physically interacts with glutamate receptors in two highly metastatic breast cancer cell lines. HGF, which creates a supportive proinvasive microenvironment for the tumor cells, stabilizes this interaction. Pharmacological inhibition of glutamate receptors blunts the migration and invasion elicited by HGF, suggesting drug repurposing of glutamate receptor antagonists for anticancer therapy.

**Abstract:**

The N-methyl-D-aspartate receptor (NMDAR) is a glutamate-gated ion channel involved in excitatory synaptic transmission. Outside the nervous system, the NMDAR is expressed in a variety of tissues and in cancers, notably in the highly invasive and metastatic triple-negative breast carcinoma. MET encodes the tyrosine kinase receptor for HGF and is a master regulator gene for “invasive growth”. In silico analysis shows that high expression of the NMDAR2B subunit is a negative prognostic factor in human invasive breast carcinoma. Here, we show that in triple-negative breast cancer cell lines NMDAR2B and MET proteins are coexpressed. HGF stimulation of these cells is followed by autophosphorylation of the MET kinase and phosphorylation of the NMDAR2B subunit at tyrosines 1252 and 1474. MET and phosphorylated NMDAR2B are physically associated, as demonstrated by co-immunoprecipitation, confocal immunofluorescence, and proximity ligation assays. Notably, pharmacological inhibition of NMDAR by MK801 and ifenprodil blunts the biological response to HGF. These results demonstrate the existence of a MET-NMDAR crosstalk driving the invasive program, paving the way for a new combinatorial therapy.

## 1. Introduction

The “N-methyl-D-aspartate receptor” (NMDAR) is an excitatory receptor for glutamate expressed on neurons and involved in synaptic transmission [1]. The NMDAR is assembled as a tetramer that differs in subunit composition. To date, seven different subunits, falling into three subfamilies, have been identified. The NMDAR1 subunits, invariably present in the tetramer and ubiquitously expressed in the brain, are encoded by the *GRIN1* gene. The NMDAR2 subunits, encoded by four distinct *GRIN2* genes (A–D), and the NMDAR3 subunits, encoded by two *GRIN3* genes (A–B), are differentially expressed throughout the brain and during development. Although NMDAR is generally thought to be a brain protein, in the last decade, the receptor was found to be present in a number of tissues, including several types of cancer [2,3,4]. Among the latter, elevated expression of *GRIN2B* was significantly associated with triple-negative breast cancer (TNBC) characterized by unfavorable prognosis, lack of expression of hormone receptors (ER and PR), and of the growth factor receptor Her2 [5]. Triple-negative breast cancers are endowed with invasive and metastatic properties.

MET is an oncogene, encoding the tyrosine kinase receptor of hepatocyte growth factor (HGF), a cytokine unleashing the invasive, and metastatic phenotype consisting of cell-cell dissociation, migration, protection from apoptosis, invasion, and cell proliferation [6,7]. This phenotype is a hallmark of triple-negative breast cancers. MET expression is particularly high in these tumors [8]. In this work, we investigated whether a functional correlation exists between the NMDAR and the MET receptor. We show that the NMDAR physically associates with the MET receptor. This association is enhanced after HGF stimulation and results in the phosphorylation of NMDAR at Tyr 1252 or 1474. Furthermore, the biological response to HGF is inhibited by MK801 or ifenprodil, two specific NMDAR inhibitors. These results suggest that the well-known pharmacology of the glutamate channel may be exploited for therapeutic inhibition of MET-induced invasive growth.

## 2. Materials and Methods

### 2.1. Bioinformatic Analysis

Gene expression analysis of the *GRIN* gene family and survival analysis were produced by web-based tools querying “The Cancer Genome Atlas” (TCGA) “omics” data (normal tissues and different cancer types). Data used for Kaplan–Meier survival analysis were obtained from the os_2904_BRCA database in The Cancer Omics Atlas (TCOA). The population was composed of patients with breast invasive carcinomas at stages II and III. All subtypes of breast carcinoma, classified on the basis of estrogen receptor (ER), progesterone receptor (PR), and human epidermal growth factor receptor 2 (HER2) expression, were included. In addition, both subclasses, luminal-A and luminal-B, were considered. Death events that occurred in the last 25 years were analyzed to build the survival curves. Kaplan–Meier analysis was performed through the GraphPad Prism v.8 software (GraphPad Software, San Diego, CA, USA), and survival was reported as a percentage. The names of abbreviations for cancer types are reported in Appendix A.

### 2.2. Cell Culture and Materials

The human breast carcinoma cell lines (BT-549, MDA-MB-231, BT-474, EFM-192A, MDA-MB-453, SK-BR-3, and ZR-75-1) were purchased from the American Type Culture Collection (ATCC, Manassas, VA, USA) and cultured in RPMI, except for MDA-MB-453 and SK-BR-3 in DMEM. The media were added with 10% of FBS, 1% of penicillin, 1% of streptomycin, and 1% of L-glutamine. Approximately 20% of FBS was used for EFM-192A and BT-474 cells. Medium for BT-474 was supplemented with 10 μg/mL of insulin. All reagents, unless specified, are from SigmaAldrich (St. Louis, MO, USA). HGF (recombinant human hepatocyte growth factor NS0-expressed) was purchased from R&D systems (Minneapolis, MN, USA). MET tyrosine kinase inhibitor JNJ-38877605 (JNJ) was purchased from Selleckchem (Houston, TX, USA).

### 2.3. Western Blot Analyses

BC cells were lysed in ice-cold RIPA added with protease inhibitor cocktail. Lysates were subsequently sonicated and centrifuged at 12,000× *g* at +4 °C for 20 min. BCA Protein Assay Kit (Thermo Fisher Scientific, Waltham, MA, USA) was used to evaluate protein concentration. Normalized protein lysates were separated by electrophoresis together with a prestained protein ladder (10–180 kDa, PageRuler, Thermo Fisher Scientific, Waltham, MA, USA) on 4–12% precast gels for SDS-PAGE (Invitrogen, Carlsbad, CA, USA). After gel running, proteins were transferred to Hybond-P pvdf membrane (Bio-Rad, Hercules, CA, USA). The membranes were blocked with 10% of BSA at room temperature, and subsequently incubated with primary antibodies (Appendix A) overnight at +4 °C, and with specific HRP-conjugated secondary antibodies (Jackson Laboratory, Bar Harbor, ME, USA) for 1 h at room temperature. ECL Prime detection kit and Image Lab software (Bio-Rad, Hercules, CA, USA) were used for protein detection and quantification, respectively. 

### 2.4. Immunohistochemical Analysis

BT-549 and MDA-MB-231 cells were used to produce 5-μm-thick serial sections cut from formalin-fixed paraffin-embedded samples, mounted on slides and treated following standard procedures. Hydrogen peroxide 3% in TBS was employed for 30 min to quench endogenous peroxidases. Antigen retrieval was performed by boiling the sections in citric acid, at pH 6 in a water bath at 95 °C for 1 h. Sections were incubated in a blocking solution (5% of normal horse serum for MET and normal goat serum for NMDAR2B in TBS-Tween-Triton) for 1 h and then incubated overnight at +4 °C with primary antibodies (Appendix A). The following day, sections were incubated for 1 h at room temperature with HRP-conjugated (Agilent Dako, Santa Clara, CA, USA) anti-Goat for MET and anti-Rabbit- for NMDAR2B staining. Peroxidase activity was developed with DAB (ImmPACT DAB, Vector, Burlingame, CA, USA). Nuclear counterstaining was performed using Hematoxylin (Bio-Optica, Milan, Italy). Stained tissue sections were dehydrated, and mounting media was added to apply the coverslips. Images were taken with Leica ICC50 microscope, and LAS AF Leica software was used for acquisition.

### 2.5. Immunofluorescence Analysis

Cells were plated in fibronectin (3 μg/mL)-coated 24-well plates, and fixed 10 min with ice-cold 4% of PBS paraformaldehyde (Santa Cruz Biotechnology, Dallas, TX, USA). A total of 0.1% of Triton X-100 and 1% of BSA were used for permeabilization and saturation, respectively. Then the cells were incubated with a primary antibody (Appendix A) and Alexa Fluor 488-conjugated anti-goat/rabbit secondary antibody (Invitrogen, Carlsbad, CA, USA) for 1 h at room temperature. TCS SP2 AOBS confocal laser-scanning microscope and LAS AF software (Leica Microsystems, Wetzlar, Germany) were used to obtain immunofluorescence images. Immunofluorescence quantification was performed using the ImageJ software. Green (number of replicates = 6) and yellow (number of replicates = 3/4) positive cells were counted and normalized on nuclei number (DAPI staining).

### 2.6. Immunoprecipitation Assay

BT-549 and MDA-MB-231 cells were lysed in ice-cold RIPA buffer in the presence of a cocktail of protease inhibitors. Total protein lysates were incubated on rotor with anti-MET home-made antibodies overnight at 4 °C and then with sepharose protein A (GE Healthcare Systems, Chicago, Illinois, USA) for 2 h at 4 °C. Incubation with sepharose protein A in the absence of antibodies was used as control. Subsequently, five washes with ice-cold RIPA buffer and elution with boiling Laemmli buffer were performed. Immunoprecipitated proteins were separated on SDS-PAGE and analyzed by western blotting (for primary antibodies see Appendix A).

### 2.7. Proximity Ligation Assay (PLA)

Cells were plated and fixed as described in the “Immunofluorescence analysis” section of Materials and Methods. For PLA, Duolink In Situ Detection Reagents Orange kit (DUO92007) was used following the manufacturer’s instructions. Anti-MET (homemade antibody), NMDAR2B (ab65783, Abcam, Cambridge, UK), and P-NMDAR2B (Tyr1252, 48-5200, Invitrogen, Carlsbad, CA, USA) antibodies were exploited. TCS SP2 AOBS confocal laser-scanning microscope) and LAS AF software (Leica Microsystems, Wetzlar, Germany) were used for fluorescence analysis. For the fluorescence quantification (number of replicates = 5), the ImageJ software was used for counting the red fluorescence and normalizing the nuclei number (DAPI staining).

### 2.8. Wound-Healing Assay

Cells were plated in 24-well plates (150,000 cells/well) and maintained in culture until confluence. Then were starved overnight with medium plus 0.5% of FBS. The monolayers were wounded with a plastic pipette. Then cells were untreated or treated for 24 or 48 h. Images of wound at the start moment and after the treatment were taken with DMRI Leica inverted microscope. Migration was quantified by evaluating the area of wound at time zero (A0) and after the treatment (Ay, y = 24/48 h). Normalization and quantification on the basis of three independent experiments were obtained. Areas were quantified with ImageJ software.

### 2.9. Invasion Assay

BT-549 and MDA-MB-231 cells were suspended in the upper compartment of the transwell chambers (8.0 μm of pore polycarbonate membrane insert, 3422) precoated with 40 μg/well of Matrigel (Corning Inc., New York, NY, USA). HGF (0.6 μM) was also added in the upper compartment of the transwell. The lower compartment of the chamber was filled with 10% of FBS medium and various treatments. After 24 h, cells on the upper side of the transwell filters were mechanically removed, while cells migrated through the membrane were fixed with 11% of glutaraldehyde and stained with 0.1% of crystal violet. Images were captured with optical microscopes (Leica Microsystems, Wetzlar, Germany) and cell number was quantified with ImageJ software. Normalization and quantification on the basis of three independent experiments were obtained.

### 2.10. Statistical Analysis

All values are expressed as the mean ± standard deviation (S.D.). Statistical analysis was performed blindly on groups with a sample size of at least 3. *T*-test was used to statistically compare two groups. The threshold *p*-value deemed to constitute statistical significance was <0.05 and only data characterized by *p*-values < 0.05 were denoted throughout the paper as results with statistical significance. In statistical analysis, all the samples were analyzed, and the outliers were not excluded. For Kaplan–Meier survival curves, statistical analysis was performed with Log-rank (Mantel–Cox) test. The data analysis and the graph design were created using the GraphPad Prism v.8 software (GraphPad Software, San Diego, CA, USA).

## 3. Results

### 3.1. GRIN2B Gene Is Highly Expressed in Cancer Cells and Is a Negative Prognostic Factor in Invasive Breast Cancer

We assessed the *GRIN* gene expression in human cancer by querying the TCGA database. The comparison of data between cancer samples and their normal tissue counterparts revealed that, among the *GRIN* gene family, *GRIN2B* exhibits the highest level in the majority of TCGA cancer subtypes (Figure 1a; Appendix A). Moreover, in breast invasive carcinomas, *GRIN2B* and *GRIN3B* are two *GRIN* genes mostly expressed (Figure 1b). We evaluated the possible association of these two genes with cancer patient survival. Tumor samples with *GRIN2B* high expression were associated with poor prognosis in invasive breast cancers (Figure 1c). On the other hand, *GRIN3B* gene expression did not show any correlation with cancer patient survival (Figure 1c).

### 3.2. MET and NMDAR2B Are Co-Expressed in Triple-Negative Breast Cancer (TNBC) Cell Lines

Various subtypes of BC cell lines, classified on the basis of estrogen receptor (ER), progesterone receptor (PR), and human epidermal growth factor receptor 2 (HER2) expression (Appendix A), were analyzed for NMDAR and MET protein levels by western blots (Figure 1d). The triple-negative cancer cell lines, BT-549 and MDA-MB-231, co-expressed the NMDAR and MET proteins at high levels (Figure 1d). In these cells, a strong staining for both MET and NMDAR2B proteins was also shown by immunohistochemistry (Figure 1e). Thus, we focused on these two TNBC cell lines to investigate the relationship between NMDAR and MET.

### 3.3. MET Activation Induces NMDAR2B Phosphorylation at Tyr 1252

BT-549 and MDA-MB-231 cells were treated with the ligand of MET, HGF (Figure 2). Both cell lines displayed MET phosphorylation on tyrosines 1234/1235 after 5 min and until 1 h of HGF treatment (P-MET, Figure 2a). The cotreatment of HGF and JNJ (JNJ-38877605), a specific MET tyrosine kinase inhibitor, completely blunted MET phosphorylation (Figure 2a). Interestingly, a marked increase in phosphorylation of NMDAR2B at Tyr1252 was observed (Figure 2a). Importantly, inhibition of the MET receptor with JNJ reduced the HGF-mediated phosphorylation, indicating that this effect was specifically exerted by MET (Figure 2a). Immunofluorescence analysis in both TNBC cell lines confirmed western blot experiments showing a significant stimulation of NMDAR2B phosphorylation on Tyr1252 after 30 min of HGF treatment (Figure 2b).

### 3.4. MET Physically Interacts with NMDAR2B Subunit after Stimulation with HGF

To elucidate the molecular crosstalk between MET and NMDAR2B, we investigated whether the two receptors physically interact. Co-immunoprecipitation analysis showed that the anti-MET antibody pulled down the NMDAR2B subunit in TNBC cell lysates, suggesting that the two receptors are physically associated (Figure 3a). Notably, treatment with HGF for 30 min led to strong co-immunoprecipitation with MET of NMDAR2B phosphorylated either at Tyr1252 or Tyr1474 (Figure 3a). Superimposable results were obtained by confocal immunofluorescence analysis (Figure 3b) and proximity ligation assays (Figure 4).

In Figure 3b, colocalization of MET and NMDAR2B or phospho-NMDAR2B was detected by merged staining (yellow signal). HGF treatment, and the following MET phosphorylation, strongly increased the formation of phosphorylated complexes between MET and NMDAR2B. The proximity ligation assay (Figure 4) confirmed the close interaction between the two molecules. At a steady state, a few clusters were detected, which strongly increased after the HGF treatment. Inhibition of MET activation with JNJ completely blocked the formation of complexes, demonstrating that the effect is specific (Figure 4). Altogether, these results show that MET and NMDAR2B physically interact, and that HGF stabilizes this interaction.

### 3.5. Pharmacological Inhibition of NMDAR Blunts Biological Responses Triggered by HGF

Since MET is a known master regulator gene for the invasive growth of cancer cells, the role of NMDAR in MET-induced migration and invasion was evaluated in TNBC cells (Figure 5 and Figure 6). The wound healing assay is an assay that measures migration and proliferation. The assay showed that the HGF treatment significantly induced cancer cell migration over the wound (Figure 5). The Matrigel invasion assay is an in vitro assay that evaluates the cell’s ability to invade the basal lamina. The assay indicated that HGF significantly increased the number of cancer cells passing through the Matrigel-coated membrane (Figure 6). Pharmacological inhibition of the NMDAR channel by MK801 completely blunted both migration and invasion elicited by HGF (Figure 5 and Figure 6). Since MK801 produces off-target effects, such as inhibition of nicotinic acetylcholine receptors, and serotonin and dopamine transporters, a specific NMDAR2B antagonist, ifenprodil (IF) was used. The specific IF compound yielded results superimposable to those obtained with MK801 (Figure 5 and Figure 6). Cancer cells treated with either inhibitor alone, without HGF, did not change migration and invasion (Figure 5 and Figure 6). Overall, these results demonstrate that NMDAR is involved in the MET-driven invasive program in TNBC cells.

## 4. Discussion

NMDA receptors are glutamate-gated ion channels crucial for neuronal communication. In the central nervous system (CNS), the NMDA receptor is known to be important for controlling synaptic plasticity and mediating learning and memory functions [1]. On the other hand, an increasing body of evidence shows that the HGF-MET axis is involved in a spectrum of neurological processes [9], including axonal growth [10], glutamatergic circuit development [11], synaptic plasticity [12,13,14], physiological learning and memory [15], and in a neuropathological syndrome, such as autism [16]. A recent report shows that NMDARs subunits, src and adaptor/scaffold proteins, which are partners of NMDAR, are part of the MET interactome in neurons [17]. Recently, we found that a crosstalk exists between MET and NMDAR to control neuronal survival [18]. Zeng et al. have shown that the glutamate NMDA receptor is crucial for metastatic colonization of the brain by breast cancer cells [5]. Increased expression of phosphorylated NMDAR2B was observed in invading cells of pancreatic cancer and under hydrodynamic pressure in vitro, suggesting a link between the long-recognized existence of high interstitial flow pressure in solid tumors and the hallmark capabilities for tumor invasion [19].

Given the established role of MET in the control of metastatic and invasive growth [7], we sought for a link between the HGF-MET axis and the NMDAR. The MET is frequently associated with breast cancers endowed with an aggressive phenotype [8,20,21] and is one of the most differentially regulated genes in triple-negative breast cancers [22,23]. We showed, by in silico analysis of the TCGA database, that NMDAR2B is the most highly expressed gene among the *GRIN* family and is associated with poor prognosis in invasive breast cancers, confirming the previous study by Zeng et al. [5]. By screening different breast cancer cell lines, we found that triple-negative breast cancers co-express high levels of MET and NMDAR2B. From co-immunoprecipitation, confocal immunofluorescence, and proximity ligation experiments, we provided direct evidence that the MET receptor tyrosine kinase physically interacts with the NMDAR2B subunit, and that the interaction is stabilized by HGF. We report that HGF-induced autophosphorylation of the MET kinase is followed by phosphorylation of the NMDAR2B subunit at tyrosines 1252 and 1474. The latter is phosphorylated by Fyn kinase in the CNS [24]. Phosphorylation of Tyr 1474 was shown to stabilize synaptic NMDAR on the cell surface by preventing the interaction of the clathrin adaptor protein with the YEKL motif, and hence endocytosis [25]. Phosphorylation of Tyr 1252 fosters the binding to the actin-regulatory protein Nck2, thus, enhancing NMDAR functions [26]. Indeed, NMDAR associates with actin and a rich network of cytoskeletal proteins, including myosin IIB, paxillin, ezrin, and cortactin [27]. These molecules are well-known downstream targets and mediators employed by MET for the dynamic regulation of cell-cell and cell-matrix adhesion, migration, and invasion [28,29,30,31,32]. We propose that HGF stabilizes the interaction between MET and NMDAR and increases phosphorylation of Tyr 1252 and 1474, coupling the channel to the cytoskeleton to regulate NMDAR activities, as suggested in neurons [33]. In this view, pharmacological inhibition of NMDAR leads to impairment of cell migration and invasion in response to HGF. Our findings unveil new roles for NMDAR in cancer and suggest that combination therapy with either pathway inhibitor may impair the invasive potential of MET-driven tumors.

## 5. Conclusions

This work unveils a new link between MET and NMDAR in breast cancers and highlights a new role of glutamate receptor in the proinvasive response to HGF.

## Figures and Tables

**Figure 1 cancers-14-04408-f001:**
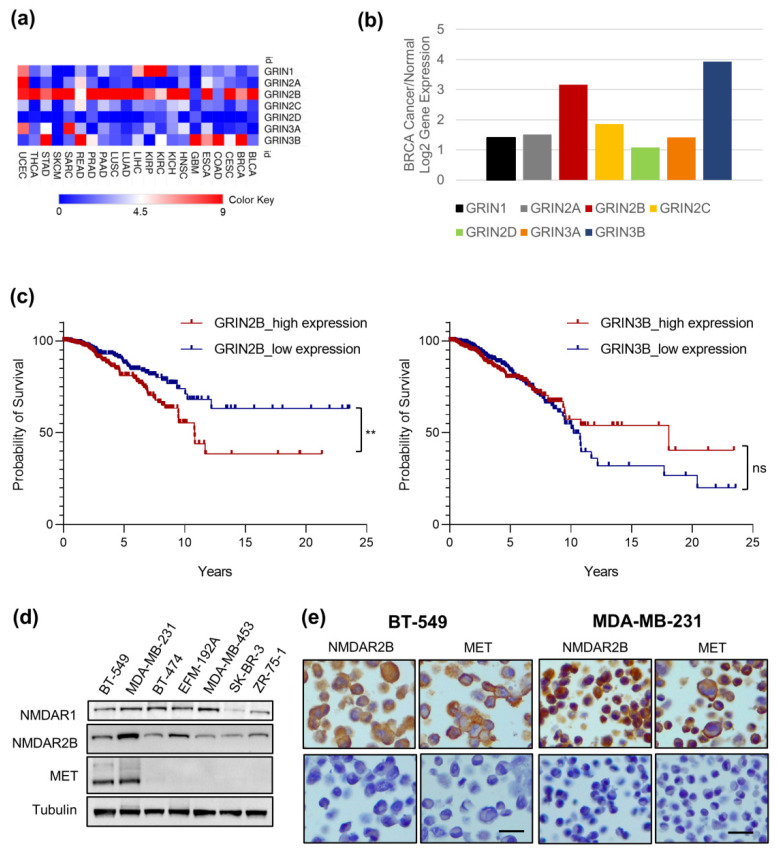
Association of NMDAR2B expression with human breast cancer. (**a**,**b**) Gene expression ratio between cancer and normal tissues for GRIN (NMDAR) gene family members using the TCGA dataset. Heatmap gene expression analysis in TCGA cancer subtypes (**a**) and representative GRIN gene expression analysis in breast invasive carcinomas (BRCA). (**b**) The names of the abbreviations for TCGA cancer subtypes are reported in Appendix A. (**c**) Survival analysis comparing samples with GRIN2B (left) and GRIN3B (right) high expression (red) and low expression (blue) in breast invasive cancer querying the TCGA database. Log-rank (Mantel–Cox) test results for GRIN2B: chi-square = 6.783, DF = 1, ** *p*-value = 0.0092; for GRIN3B: chi-square = 0.0001764, DF = 1, ^ns^
*p*-value = 0.9894. (**d**) Protein levels of NMDAR1, NMDAR2B, and MET were evaluated by western blot in different BC cell lines. Tubulin was used as loading control. Specific features for each cell line are reported in Appendix A. Original blots see Appendix A. (**e**) Immunohistochemical staining (brown color) for NMDAR2B, and MET was performed in TNBC cell lines: BT-549 and MDA-MB-231. Nuclei are counterstained with hematoxylin (blue color). Bar = 100 µm.

**Figure 2 cancers-14-04408-f002:**
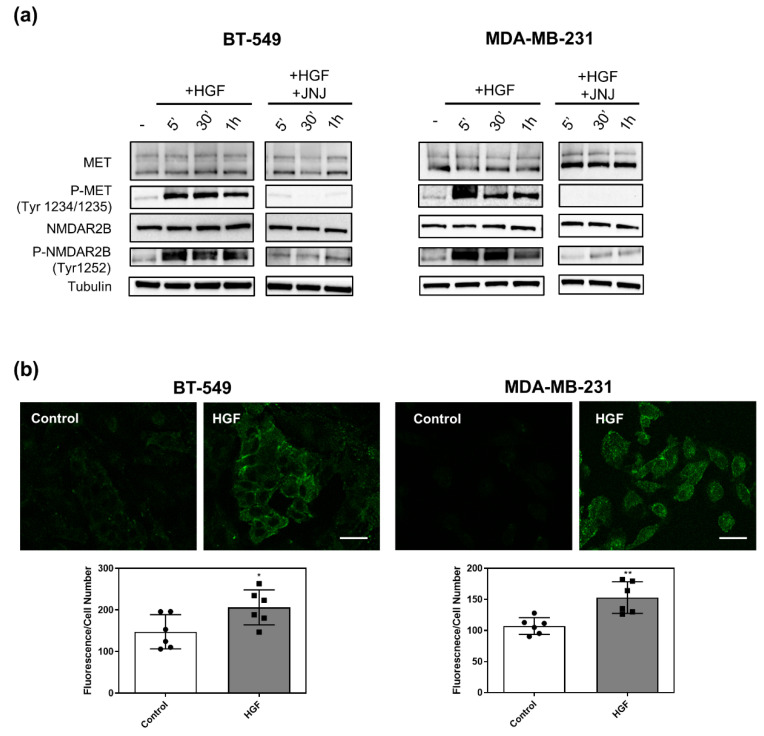
MET activation induces NMDAR2B phosphorylation at Tyr 1252. (**a**) BT-549 (left) and MDA-MB-231 (right) TNBC cells were unstimulated (-), stimulated with HGF (0.6 nM) and HGF + JNJ (JNJ-38877605, a specific MET inhibitor, 500 nM, for different lengths of time. Protein levels of total and phosphorylated MET (Tyr1234/1235) and NMDAR2B (Tyr1252) were measured by western blot. Tubulin was used as loading control. Original blots see Appendix A. (**b**) Phosphorylated NMDAR2B (Tyr1252) protein levels were measured by immunofluorescence in BT-549 (left) and MDA-MB-231 (right) cells untreated (control) or treated with HGF (0.6 nM for 30 min). Representative images (upper panels) and fluorescence quantification (lower panels) are presented. Bar = 35 µm. Values are the mean ± S.D. of six independent experiments. A *t*-test was applied for comparison of each sample versus HGF-treated cells. ** *p*-value < 0.01; * *p*-value < 0.05.

**Figure 3 cancers-14-04408-f003:**
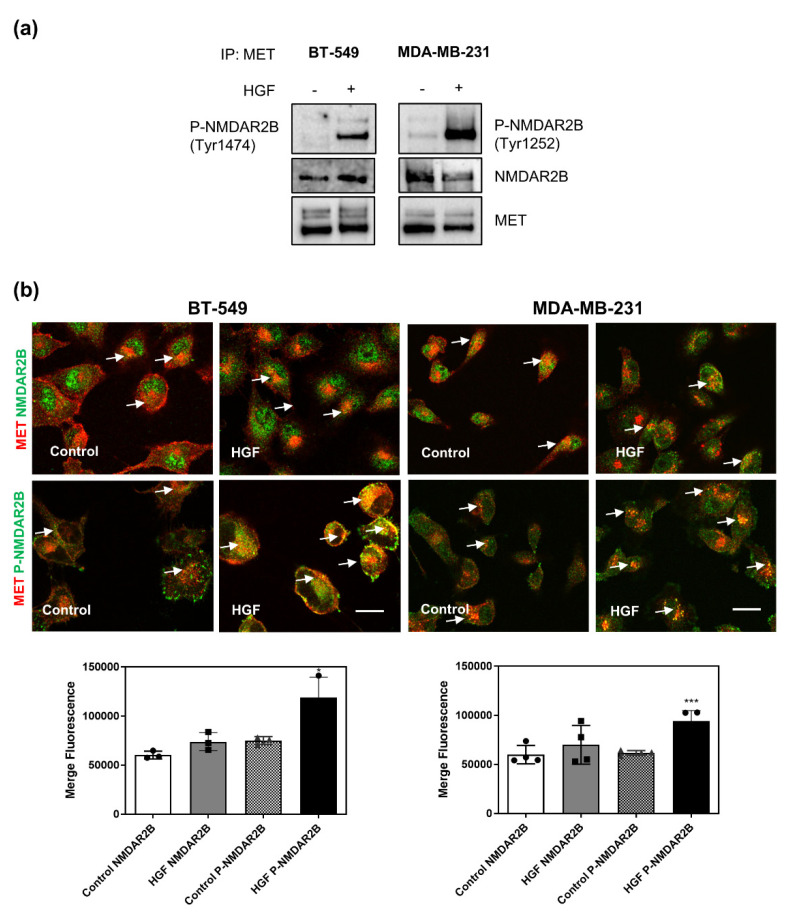
MET physically interacts with NMDAR2B after stimulation with HGF. BT-549 (left panels) and MDA-MB-231 (right panels) TNBC cells were untreated (control) or treated with HGF (0.6 nM) for 30 min. (**a**) MET IP was analyzed by western blot with phosphorylated NMDAR2B (Tyr1252 or Tyr1474), total NMDAR2B, and total MET antibodies. Original blots see Appendix A. (**b**) MET and NMDAR2B immunofluorescence co-staining (yellow, white arrows) were produced for MET (red) with total (upper panels) or phosphorylated (Tyr1252, lower panels) NMDAR2B (green) proteins. Representative confocal microscopy images of fluorescence and quantitative analysis of MET/NMDAR signals are shown. Bar = 50 µm. Values are the mean ± S.D. of three/four independent experiments. A *t*-test was applied for comparison of each sample versus HGF-treated cells. *** *p*-value < 0.001; * *p*-value < 0.05.

**Figure 4 cancers-14-04408-f004:**
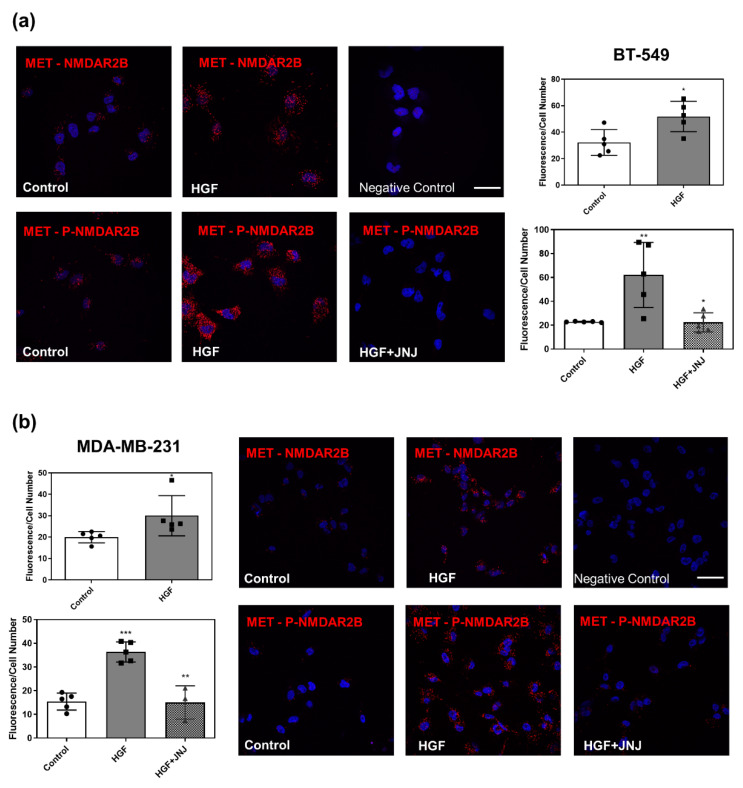
MET forms complexes with NMDAR2B after stimulation with HGF. BT-549 (**a**) and MDA-MB-231 (**b**) TNBC cells were untreated (control) or treated with HGF (0.6 nM) and HGF + JNJ (JNJ-38877605, a specific MET inhibitor, 500 nM) for 30 min. Proximity ligation assay (PLA) was performed using antibodies recognizing MET and total (upper panels) or phosphorylated (Tyr1252, lower panels) NMDAR2B proteins. Red fluorescent profiles represent regions of PLA signal amplification, denoting MET and NMDAR colocalization. Representative confocal microscopy images of PLA fluorescence (red) and quantitative analysis of MET/NMDAR PLA signals were reported. In control experiments, no PLA signal was detected using each antibody alone (negative control). Bar = 50 µm. Values are the mean ± S.D. of five independent experiments. A *t*-test was used to calculate each sample versus HGF-treated cells. *** *p*-value < 0.001; ** *p*-value < 0.01; * *p*-value < 0.05.

**Figure 5 cancers-14-04408-f005:**
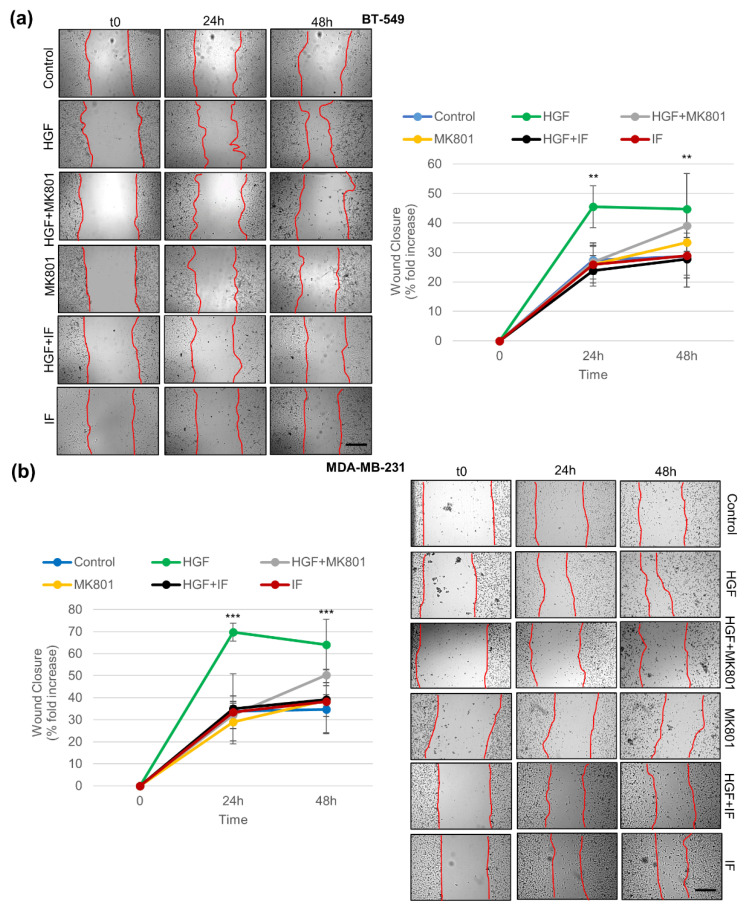
The HGF-mediated promigratory function requires NMDAR2B. BT-549 (**a**) and MDA-MB-231 (**b**) TNBC cells were untreated (control) or treated for 24 or 48 h with HGF (0.6 nM) and HGF + two different NMDAR inhibitors: MK801 (NMDAR inhibitor, 100 μM) and ifenprodil (IF, a specific NMDAR2B inhibitor, 1 μM). The cells were also treated with the two inhibitors alone as a control. The HGF-mediated promigratory function was assayed with wound healing assays. Representative images and wound closure quantification are reported. Bar = 300 µm. Migration was quantified by evaluating the area of the wound at time zero (A0) and after the treatment (Ay, y = 24 or 48 h). Values are the mean ± S.D. of three independent experiments. A *t*-test was applied to compare each sample versus HGF-treated cells. *** *p*-value < 0.001; ** *p*-value < 0.01.

**Figure 6 cancers-14-04408-f006:**
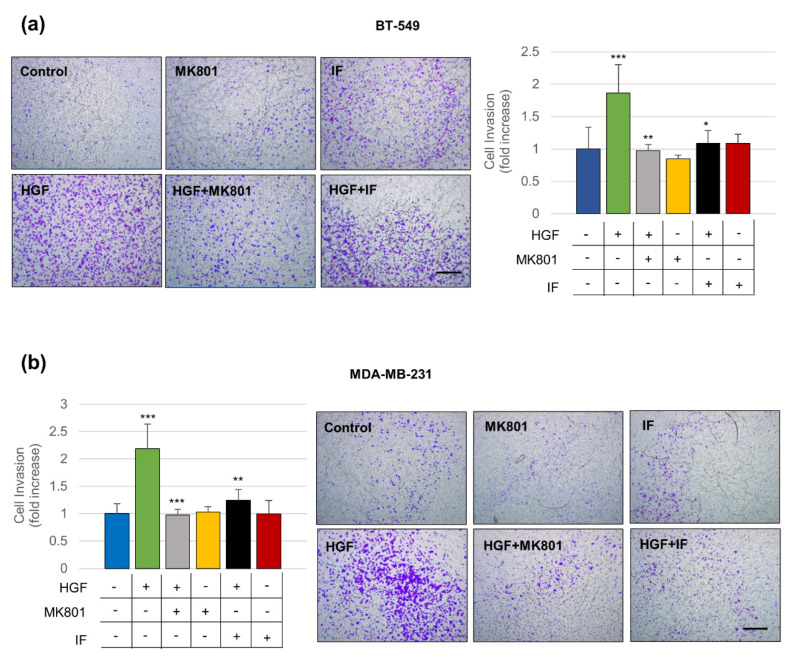
The HGF-mediated proinvasive function requires NMDAR2B. BT-549 (**a**) and MDA-MB-231 (**b**) TNBC cells were untreated (control) or treated for 24 h with HGF (0.6 nM) and HGF + two different NMDAR inhibitors: MK801 (NMDAR inhibitor, 100 μM) and ifenprodil (IF, a specific NMDAR2B inhibitor, 1 μM). The cells were also treated with the two inhibitors alone as a control. The HGF-mediated proinvasive function was assayed with the transwell invasion Matrigel assay. Representative images and invasiveness quantification are shown. Invasion was evaluated through crystal violet coloration, which correlates with the cell number. Bar = 300 µm. Values are the mean ± S.D. of three independent experiments and are expressed as ‘fold mean control’. A *t*-test was applied to compare each sample versus HGF-treated cells. *** *p*-value < 0.001; ** *p*-value < 0.01; * *p*-value < 0.05.

## Data Availability

The data generated in this study are available within the article and its Appendix A. Data used for gene expression analysis of the *GRIN* gene family and survival analysis were obtained from The Cancer Omics Atlas (TCOA) at os_2904_BRCA.

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
