# Peer review of "MET Oncogene Controls Invasive Growth by Coupling with NMDA Receptor"

_cancers, 2022, doi:10.3390/cancers14184408_

Round 1
Reviewer 1 Report
This manuscript reported a new link between MET and NMDAR using two triple negative breast cancer cell lines and different approaches, suggesting a new role of glutamate receptor in the pro-invasive response to HGF. The manuscript is well drafted, and data are clearly presented.
Minor
MK801 also inhibits nicotinic acetylcholine receptor, serotonin and dopamine transporters. Using 100µM in the manuscript may cause some non-specific effects. This should be considered in the result interpretation as NMDAR knockout experiments or siRNA knockdown was not performed.
Figure 2. Based on the results, MET physically interacting with NMDAR2B after stimulation with HGF might be more accurate?
Some spelling, e.g. The "N-methyl-D-aspartate receptor" (NMDAR) is an excitatory receptor for gluta-36 mate expressed on neurons and involved in synaptic transmission [1] – missing full stop after the end of sentence.
Author Response
1) MK801 also inhibits nicotinic acetylcholine receptor, serotonin and dopamine transporters. Using 100μM in the manuscript may cause some non-specific effects. This should be considered in the result interpretation as NMDAR knockout experiments or siRNA knockdown was not performed.
We thank the Reviewer to have done this consideration regarding the non-specific effect exerted by the MK801 NMDAR inhibitor. We agree that MK801 acts on off-targets, such as nicotinic acetylcholine receptor, and serotonin and dopamine transporters. For this reason, we used the NMDAR2B specific inhibitor ifenprodil in parallel with MK801 (Figure 5 and 6). Importantly, we found superimposable results between the two inhibitors. Following the reviewer’ suggestion we added this consideration in the results section (3.5 paragraph) of the revised manuscript.
2) Figure 2. Based on the results, MET physically interacting with NMDAR2B after stimulation with HGF might be more accurate?
We agree with the Reviewer, that MET interacts with NMDAR2B mainly after stimulation with HGF. Following his/her suggestion, the new title of 3.4 Results paragraph is “MET physically interacts with NMDAR2B subunit after stimulation with HGF”. Accordingly, we changed also the title in the legends to Figure 3 and 4.
3) Some spelling, e.g. The "N-methyl-D-aspartate receptor"(NMDAR) is an excitatory receptor for gluta-36 mate expressed on neurons and involved in synaptic transmission [1] – missing full stop after the end of sentence.
We thank the Reviewer to have highlighted this spelling error, we added the missing full stop in the revised manuscript.
Reviewer 2 Report
Investigators present novel data indicating a physical interaction between MET and NMDAR in triple negative breast cancer cells. It was demonstrated that NMDAR mediates, or partially mediates, responses to HGF stimulation of MET, as inhibiting NMDAR with pharmacological agents was shown to impair HGF-induced invasion and migration of TNBC cell lines. These cell line data support the observations from clinical datasets, that higher expression of NMDAR2B (GRIN2B) associates with poorer clinical outcomes in breast cancer.
The study would be greatly strengthened with additional experiments to show that siRNA knockdown or CRISPR knockout of NMDAR phenocopies the pharmacological inhibition of NMDAR in TNBC cell lines. Additionally, demonstration that ectopic expression of MET in NMDAR2B+ cells (such as BT474 or SKBR3) enables HGF-induced invasion and migration, and that such invasion and migration is then inhibited by NMDAR inhibitors, would be beneficial.
The data is presented well, but some additional details are needed. Specific examples follow:
Figure 1c: details regarding the patient population must be presented.
Figure 1c: results of the logrank test or of other appropriate test must be presented for the Kaplan Meier survival curves.
Figure 1e: staining of MET and NMDMAR2B in MET-negative cell line, such as BT474 or SKBR3, should be presented for comparison.
Author Response
1) The study would be greatly strengthened with additional experiments to show that siRNA knockdown or CRISPR knockout of NMDAR phenocopies the pharmacological inhibition of NMDAR in TNBC cell lines.
We thank the Referee to have suggested new additional experiments to improve our manuscript. Certainly, the knockdown experiment would be a further confirmation of our data. However, these experiments would take well over the five days allowed by the Editor for the revision of the manuscript. To demonstrate that the pharmacological inhibition of NMDAR blunts biological responses triggered by HGF, we have used two different inhibitors of NMDAR: MK801 and ifenprodil (IF). Since MK801 may act on off-targets, such as nicotinic acetylcholine receptor, and serotonin and dopamine transporters, we also used, in parallel with MK801, the highly specific NMDAR2B inhibitor, ifenprodil (Figure 5 and 6). Importantly, we found superimposable results by using the two inhibitors. We think that our results support the conclusion proposed in the manuscript.
Additionally, demonstration that ectopic expression of MET in NMDAR2B+ cells (such as BT474or SKBR3) enables HGF-induced invasion and migration, and that such invasion and migration is then inhibited by NMDAR inhibitors, would be beneficial.
This further experimental approach could be very interesting and will be surely included in a future study extended to MET-negative breast cancer cells, such as BT474 and SKBR3. We believe that these further experiments are beyond the scope of this work, which focused on triple negative breast cancer cells.
2) The data is presented well, but some additional details are needed. Specific examples follow:
Figure 1c: details regarding the patient population must be presented.
According with Referee’s comment, details regarding the patient population were added in the revised manuscript in Materials and Methods section at “2.1. Bioinformatic Analysis” paragraph.
Figure 1c: results of the log rank test or of other appropriate test must be presented for the Kaplan Meier survival curves.
We thank the reviewer for suggesting the appropriate statistical analysis for Kaplan Meier survival curves. Following his/her comment we inserted the statistical results of the Log-rank (Mantel-Cox) test in the new Figure 1 and relative legend.
Figure 1e: staining of MET and NMDMAR2B in MET-negative cell line, such as BT474 or SKBR3, should be presented for comparison.
We thank the Reviewer for his/her comment. The work was designed to determine the interaction between MET and NMDAR2B in breast cancer, thus we focused on MET-positive cells. In literature it has been reported that BT474 or SKBR3 cell lines do not express MET at detectable levels. For a reference You can see Figure 1 in Stanley A. et al. “Synergistic effects of various Her inhibitors in combination with IGF-1R, C-MET and Src targeting agents in breast cancer cell lines” Scientific Reports 2017, 7, 3964. In our view, we feel that this comparison is not essential for our main results on triple negative breast cancer cells.